# Decreasing Tacrolimus Concentrations in Routine Therapeutic Drug Monitoring Data Indicate Adherence to Updated Therapeutic Goals

**DOI:** 10.3390/biomedicines14010094

**Published:** 2026-01-02

**Authors:** Anders Larsson, Johan Saldeen, Jonathan Cedernaes, Mats B. Eriksson, Mathias Karlsson, Anna-Karin Hamberg

**Affiliations:** 1Department of Medical Sciences, Clinical Chemistry and Pharmacology, Uppsala University, SE-751 85 Uppsala, Sweden; johan.saldeen@akademiska.se (J.S.); jonathan.cedernaes@akademiska.se (J.C.); mathias.karlsson@capitainer.se (M.K.);; 2Department of Surgical Sciences, Uppsala University, SE-751 85 Uppsala, Sweden; mats.b.eriksson@uu.se; 3NOVA Medical School, New University of Lisbon, 1099-085 Lisbon, Portugal

**Keywords:** tacrolimus, patient median, therapeutic drug monitoring, immunosuppressive, transplantation

## Abstract

**Background:** Tacrolimus is a key immunosuppressive drug used to prevent organ rejection after transplantation. Its narrow therapeutic window and high interindividual pharmacokinetic variability make therapeutic drug monitoring (TDM) essential. This study aimed to (1) characterize long-term trends in tacrolimus concentrations; (2) assess potential seasonal variation; and (3) evaluate the suitability of patient medians as a tool for laboratory quality monitoring. **Methods:** A retrospective analysis was conducted on 113,735 tacrolimus whole-blood results obtained between 2006 and 2024 at Uppsala University Hospital, Sweden. Samples were analyzed using immunoassays on the Hitachi 912 (Microgenics) until 2008, the Abbott Architect until 2021, and the Roche Cobas Pro e 801 thereafter. Annual patient medians and percentiles (10th, 25th, 50th, 75th, and 90th) were calculated. Seasonal variation was assessed by comparing monthly test volumes and concentration distributions. **Results:** The annual number of tacrolimus results increased from 5616 in 2006 to 7320 in 2024, comprising 67,133 male and 46,602 female patient results. All distribution metrics declined steadily over the study period, with median tacrolimus concentrations decreasing by 20–30%. The July test volume was approximately 15% lower than in other months, but no meaningful seasonal variation in tacrolimus concentrations was observed; monthly medians and percentiles remained stable throughout the year. **Conclusions:** Tacrolimus concentrations at the population level have decreased consistently over nearly two decades. These findings likely reflect evolving clinical practice, including dose minimization strategies to reduce toxicity. Patient medians proved robust over time and may serve as a useful adjunct to conventional quality control, particularly when commercial control materials do not fully mimic patient sample behavior.

## 1. Introduction

Tacrolimus is an immunosuppressive agent widely used to prevent organ rejection following transplantation [1]. Its primary mechanism of action is the inhibition of T-lymphocyte activation [2], a critical process in graft rejection. Tacrolimus is routinely prescribed in kidney, liver, heart, and lung transplants [3], typically in combination with other immunosuppressants such as mycophenolate mofetil and corticosteroids.

Tacrolimus binds to the FK506-binding protein (FKBP12), forming a complex that inhibits calcineurin, an enzyme essential for T-cell activation [4]. This inhibition prevents the dephosphorylation of the nuclear factor of activated T cells (NFAT), thereby blocking its translocation to the nucleus. As a result, transcription of interleukin-2 (IL-2) and other cytokines required for T-cell proliferation and activation is suppressed [5]. By dampening the immune response, tacrolimus reduces the risk of both acute and chronic rejection. However, its narrow therapeutic window necessitates careful dose adjustment and regular monitoring to balance efficacy with toxicity. Excessive exposure can lead to nephrotoxicity, neurotoxicity, hypertension, or diabetes, whereas insufficient levels increase the risk of rejection [6].

Therapeutic drug monitoring (TDM) is therefore central to the clinical management of transplant patients receiving tacrolimus. TDM involves measuring whole blood through concentrations (C_0_), typically collected immediately before the next scheduled dose [7]. Target ranges vary depending on the transplanted organ, time since transplantation, and concomitant therapy [8]. Individualized dosing is particularly important in the early post-transplant period when fluctuations in drug exposure are most pronounced [9,10].

Tacrolimus pharmacokinetics are highly variable among individuals. The drug is primarily metabolized by cytochrome P450 3A4 and 3A5 enzymes in the liver and intestine [11,12,13]. Genetic polymorphisms in CYP3A5 are well-established determinants of tacrolimus clearance: CYP3A5 expressers (e.g., carriers of the CYP3A5*1 allele) generally require higher doses to reach target concentrations compared with non-expressers [13]. Other factors—including age, hematocrit, drug interactions, and liver function—also influence metabolism and distribution. For instance, CYP3A inhibitors such as azole antifungals or macrolide antibiotics markedly increase tacrolimus concentrations, while enzyme inducers such as rifampicin or certain anticonvulsants reduce exposure [14,15].

Tacrolimus exhibits nonlinear pharmacokinetics, meaning small changes in dose or metabolic capacity can cause disproportionately large shifts in blood concentration [15]. Consequently, standardized dosing protocols are often inadequate, and personalized titration guided by TDM is essential. Moreover, intra-individual variability in tacrolimus levels has been associated with poorer graft outcomes, underscoring the importance of maintaining stable drug exposure [16].

Accurate measurement of tacrolimus requires highly specific and sensitive analytical methods. Historically, immunoassays such as microparticle enzyme immunoassay (MEIA), chemiluminescent microparticle immunoassay (CMIA), and electrochemiluminescence immunoassay (ECLIA) have been widely used in clinical laboratories due to their automation and convenience [17,18]. However, these assays may suffer from cross-reactivity with tacrolimus metabolites, leading to overestimation of parent drug concentrations. Liquid chromatography–tandem mass spectrometry (LC–MS/MS) offers superior specificity but is less commonly adopted for routine testing due to higher costs and increased technical demands [19].

In Sweden, routine tacrolimus measurement has been performed in clinical laboratories since the early 2000s, supporting both patient management and analytical quality assurance. Over time, changes in analytical platforms, calibration systems, and clinical guidelines have influenced result interpretation. For example, target trough levels have gradually been lowered in many transplant protocols to reduce toxicity while maintaining adequate immunosuppression [20,21]. Such adjustments may be reflected in long-term laboratory data as gradual shifts in population-level tacrolimus concentrations.

Given the central role of tacrolimus monitoring in transplant medicine, there is increasing interest in using aggregated laboratory data to assess analytical performance and clinical trends. One promising approach is the use of patient medians—the median of all patient results within a defined period—as a complement to traditional quality control. Unlike synthetic control materials, which may not fully replicate patient sample characteristics, patient medians directly reflect the analytical process and its interaction with clinical practice. This strategy can help identify calibration shifts, assay drift, or broader changes in clinical management that might otherwise remain undetected.

The aim of this study was to investigate long-term trends in tacrolimus concentrations measured at Uppsala University Hospital between 2006 and 2024. Specifically, we sought to (1) analyze yearly and seasonal variations in tacrolimus results; (2) evaluate patient medians as indicators of analytical stability; and (3) explore the implications of observed changes for both laboratory quality monitoring and clinical practice.

## 2. Materials and Methods

### 2.1. Samples

Routine requests for tacrolimus blood testing at the Departments of Clinical Chemistry and Pharmacology, Akademiska Hospital, Uppsala, were collected in EDTA tubes (367862, BD Vacutainer Systems, Plymouth, UK). The study period extended from January 2006 to December 2024 (n = 113,735). Ethical approval was obtained from the Uppsala University Ethics Committee (Dnr 01-367). Data were extracted in accordance with the ethical approval, excluding full patient identities. Only sampling date, patient age (in years), sex, and tacrolimus concentrations were included. Because the results were reported only at the group level, there was no risk of identifying individual patients, and informed consent was therefore not required by the ethics committee. Furthermore, all testing was performed as part of the patients’ routine clinical workup, meaning that no additional blood sampling was necessary for the study.

### 2.2. Instruments

Tacrolimus was initially measured using a Hitachi 912 analyzer (Roche Diagnostics, Rotkreuz, Switzerland) with Microgenics reagents (Microgenics, Fremont, CA, USA). In February 2008, the method was transferred to the Architect platform (Abbott Laboratories, Abbott Park, IL, USA) using reagent 1L77 (Abbott Laboratories). Calibration was performed with 1L77-01 (Abbott Laboratories). Internal quality control material (1P05-10) was obtained from Bio-Rad Laboratories (Hercules, CA, USA), and external controls were provided monthly by UK NEQAS (Sheffield, UK). The total coefficient of variation (CV) was 6.2% at 6.6 µg/L and 4.6% at 25 µg/L.

In January 2021, the tacrolimus assay was transferred from the Architect platform to the Cobas Pro e 801 platform (Roche Diagnostics) using tacrolimus reagents from the same manufacturer (cat. no. 07251254190). Internal controls were supplied by More Diagnostics (Los Osos, CA, USA), and external quality controls were provided monthly by LGC Standards (Wesel, Germany). The total CV was 3.5% at 5.0 µg/L and 3.6% at 20.8 µg/L.

### 2.3. Statistical Calculations

Patient medians were calculated based on yearly tacrolimus results. Statistical analyses were performed using Excel 365 (Version 2502,Microsoft Corporation, Seattle, WA, USA) and Statistica 10 (TIBCO Software, Palo Alto, CA, USA).

## 3. Results

### 3.1. Changes in Number of Reported Tacrolimus Results over Time

During 2006–2024, a total of 113,735 Tacrolimus results were reported. 67,133 of the results were for male patients, and 46,602 were for female patients. The number of test results increased during the study period from 5616 in 2006 to 7320 in 2024 (Figure 1).

The median age of the female study subjects was 52 years (interquartile range (IQR) 39–63 years) at the time of their testing. The median age of the male study subjects was 52 years (IQR 42–66 years) at the time of their testing.

### 3.2. Changes in Reported Tacrolimus Results over Time

The 10th percentile, lower quartile, median, upper quartile and 90th percentile were compared over time. There was a steady decline for all five levels over time (Figure 2). The decrease was 20–30% over the entire study period.

The mean tacrolimus value for females was 7.99 µg/L (IQR 5.60–9.60). The female test results showed a significant decrease in tacrolimus values over time (Spearman’s rank −0.214; *p* < 0.00000000001). There was also a very weak positive association between the age of the females and tacrolimus results (Spearman’s rank 0.030; *p* < 0.00000001). The male study subjects had a mean tacrolimus value of 7.96 µg/L (IQR 5.60–9.60). The male test results showed a significant decrease in tacrolimus values over time (Spearman’s rank −0.190; *p* < 0.00000000001). There was also a very weak positive association between the age of the males and tacrolimus values (Spearman’s rank 0.058; *p* < 0.00000000001).

### 3.3. Seasonal Variation in Tacrolimus Results

July is the main vacation month in Sweden. The number of tacrolimus results in July was 8041, compared with 19,299 in May and September combined, indicating a substantial reduction in test volume in July. Importantly, this reduction had no meaningful impact on the distribution of results, as the median tacrolimus concentration remained essentially unchanged: 7.2 µg/L in July versus 7.1 µg/L across May and September. This is a significant reduction calculated by Mann–Whitney U test (*p* = 0.028). However, the reduction in number of test results only increased the median tacrolimus results by 1.4% (Figure 3 and Figure 4).

### 3.4. Method Comparison Between Architect and Cobas Methods in 2021

Method comparison with fresh patient samples between the Architect and Cobas methods during the transfer of the tacrolimus method from the Architect platform to the Cobas platform in 2021 showed small differences. A mean difference of −0.23 µg/L (−1%) was considered acceptable for a method transfer with unchanged target values.

## 4. Discussion

This study presents a comprehensive longitudinal analysis of tacrolimus blood concentrations in a large transplant population over a 19-year period. With more than 113,000 individual results, the dataset provides a unique overview of both analytical and clinical trends across nearly two decades. The findings reveal a clear and continuous decline in tacrolimus concentrations across all distribution percentiles—from the 10th to the 90th—despite an overall increase in the annual number of tests performed. These results offer valuable insights into evolving clinical practice, assay standardization, and the optimization of immunosuppressive management in transplant recipients.

### 4.1. Long-Term Decline in Tacrolimus Concentrations

The observed downward trend in tacrolimus concentrations aligns with the gradual evolution of post-transplant immunosuppressive strategies. Early protocols in the 1990s often employed relatively high-target trough concentrations to prevent acute rejection. However, increasing clinical experience and long-term outcome data have shifted practice toward minimizing exposure in stable patients to reduce adverse effects such as nephrotoxicity, neurotoxicity, and metabolic complications [20,22,23].

Large-scale studies and international consensus guidelines have progressively recommended lower target concentrations, particularly beyond the first six months after transplantation. For kidney transplant recipients, typical targets have shifted from 8 to 12 µg/L in the early post-transplant phase to 4–7 µg/L, or even lower, during maintenance therapy. Similar reductions have been adopted in liver, heart, and lung transplant protocols, with individualized targets now emphasized over fixed ranges. The overall decline in median tacrolimus levels observed here likely reflects the implementation of these recommendations and the broader clinical trend toward precision dosing guided by patient-specific risk assessment.

### 4.2. Clinical and Pharmacological Drivers

The increasing use of combination immunosuppressive regimens has also contributed to lower tacrolimus exposure. Modern management often employs reduced tacrolimus doses alongside adjunct agents such as mycophenolate mofetil, azathioprine, or mTOR inhibitors (e.g., everolimus, sirolimus). These combinations provide synergistic immunosuppression while mitigating the toxicity associated with high tacrolimus levels [24].

Advances in patient monitoring, adherence programs, and pharmacogenetics have further supported more stable and lower therapeutic concentrations. Recognition of CYP3A5 polymorphisms as key determinants of tacrolimus metabolism has enabled genotype-guided dosing in some centers [25,26,27]. CYP3A5 expressers, who metabolize tacrolimus more rapidly, may receive higher initial doses, whereas non-expressers require lower doses to avoid overexposure. Adoption of genotype-guided dosing likely contributed to reduced inter-individual variability and the gradual decline in median concentrations over time.

Although the decline is primarily attributed to clinical practice changes, analytical factors must also be considered. During the study period, tacrolimus assays transitioned from the Microgenics reagent on the Hitachi 912 platform to the Abbott Architect CMIA method, and later to the Roche Cobas Pro e801 platform. Each platform employs different immunoassay designs, with inherent differences in calibration, antibody specificity, and potential cross-reactivity with metabolites. Before each transit, the new method was validated to ensure equal test results. This validation included testing of patient samples with both old and new methods to ensure compatible test results over time. The mean bias when transferring the tacrolimus method from Architect to Roche was 0.23 µg/L, which was considered acceptable (Figure 5).

However, several observations suggest that these transitions did not drive the downward trend. All method transfers were validated with overlapping parallel measurements and continuous participation in external quality assessment schemes (UK NEQAS, LGC Standards). No abrupt discontinuities or stepwise shifts corresponding to method changes were evident, and both internal and external quality controls remained within acceptable limits (CV 3–6%). The gradual, continuous nature of the decline supports a clinical rather than analytical explanation.

It is important to note that immunoassays may differ from LC–MS/MS methods in absolute values due to metabolite cross-reactivity. In this context, patient medians serve as a valuable internal benchmarking tool. A method change or calibration issue would typically produce a stepwise deviation in patient medians, independent of clinical shifts. The absence of such discontinuities reinforces the reliability of the present findings.

### 4.3. Seasonal Patterns and Healthcare Organization

A notable observation was the seasonal reduction in the number of tacrolimus analyses, particularly in July, coinciding with the main vacation period in Sweden. Outpatient visits and elective procedures typically decline during this time. Despite a ~15% reduction in test volume, median tacrolimus concentrations remained stable, indicating that dose adjustments and monitoring practices are robust even during periods of reduced clinical activity. This stability likely reflects well-structured outpatient follow-up systems and patient self-management routines. In the UK, there is a seasonal variance in renal transplantation, peaking during late autumn, and the lowest frequency of transplant per day was noted during the late summer [28].

The absence of seasonal fluctuations in concentration levels also suggests that environmental or physiological factors such as temperature, diet, or sun exposure have a limited impact on tacrolimus pharmacokinetics at the population level. This contrasts with earlier reports suggesting modest seasonal variation in drug metabolism, potentially linked to vitamin D-mediated CYP3A4 induction [29,30,31]. However, the effect size of such phenomena appears negligible in clinical practice when consistent TDM protocols are applied.

A key conceptual contribution of this study is the use of patient medians as an adjunct to conventional quality control. Traditional QC materials are essential for daily assay validation but may not fully capture long-term analytical drifts or differences in sample commutability. Patient medians, derived from large and stable test volumes, provide a real-world reference that reflects both analytical and clinical factors. Monitoring patient medians over time enables laboratories to detect gradual calibration shifts or reagent lot effects that might otherwise go unnoticed. We recently showed a clear shift in the Roche folate method using patient median values [32], supporting the use of patient medians to detect analytical drift.

In this study, patient medians behaved consistently and predictably, paralleling other percentiles and confirming their suitability as a robust internal quality indicator. This approach is particularly advantageous for tacrolimus assays, where QC materials are often matrix-modified and may not perfectly mimic patient whole blood samples. Incorporating patient median monitoring could therefore strengthen laboratory quality assurance systems, complementing external proficiency testing and method validation programs.

From a clinical perspective, the steady decline in tacrolimus concentrations suggests that immunosuppressive management has become increasingly precise and cautious. Lower exposure levels without apparent compromise in graft survival imply that clinicians have successfully balanced efficacy with toxicity. This trend reflects the global movement toward individualized immunosuppression and underscores the central role of TDM in optimizing patient outcomes.

From a laboratory perspective, the findings highlight the importance of maintaining continuous, high-quality analytical performance across changing technologies. The ability to generate harmonized data spanning nearly two decades demonstrates the maturity of tacrolimus measurement as a clinical service. Moreover, this study illustrates the potential of leveraging accumulated laboratory data for research, surveillance, and quality improvement—transforming routine test results into a valuable resource for translational medicine.

Several limitations must be acknowledged. The most significant is the absence of clinical metadata such as transplant type, time since transplantation, and concomitant therapy. Without this information, changes in concentration distributions cannot be directly attributed to specific patient subgroups or treatment modifications. Similarly, the lack of information on whether samples were drawn as true trough levels limits the precision of pharmacokinetic interpretation. These limitations, inherent to retrospective laboratory-based studies, highlight the need for integrating laboratory information systems with clinical registries in future research.

Another limitation is the exclusive reliance on tacrolimus immunoassay data. Although widely used in hospital laboratories, immunoassays may differ from LC–MS/MS methods in absolute values. Future studies incorporating mass spectrometry-based reference methods could provide further validation of the long-term trends observed here.

Finally, while patient medians offer valuable aggregate information, they cannot fully distinguish between analytical and clinical sources of variation without complementary data. Establishing standardized frameworks for integrating patient median monitoring into laboratory quality systems—potentially with automated alert thresholds—would be an important next step.

## 5. Conclusions

This long-term study demonstrates a steady decline in tacrolimus concentrations across a large transplant population over 19 years. The trend likely reflects improved clinical management and evolving therapeutic strategies aimed at minimizing drug-related toxicity. Analytical transitions between assay platforms were well controlled and did not introduce detectable discontinuities. Patient medians proved to be a practical and informative tool for long-term quality monitoring, capable of capturing gradual analytical or clinical changes at the population level.

Overall, these findings highlight the dynamic interplay between laboratory data and clinical practice. They underscore the value of large-scale, real-world laboratory datasets for understanding trends in patient care and for optimizing both analytical quality and therapeutic outcomes.

## Figures and Tables

**Figure 1 biomedicines-14-00094-f001:**
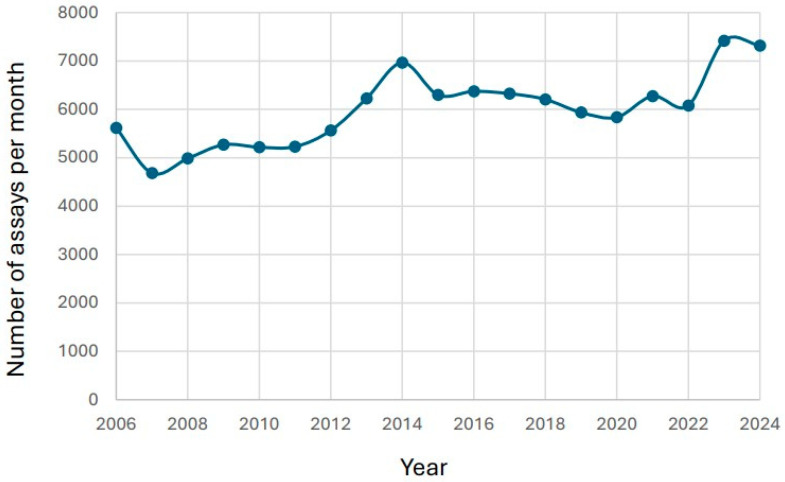
Number of tacrolimus assays/year. The data is presented for years 2006–2024.

**Figure 2 biomedicines-14-00094-f002:**
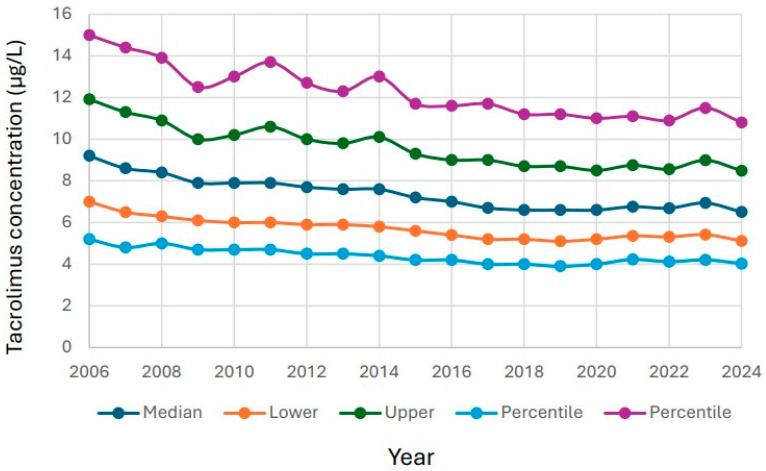
Tacrolimus results over time. The data is presented as 10th percentile, lower quartile, median, upper quartile and 90th percentile per year from 2006 to 2024.

**Figure 3 biomedicines-14-00094-f003:**
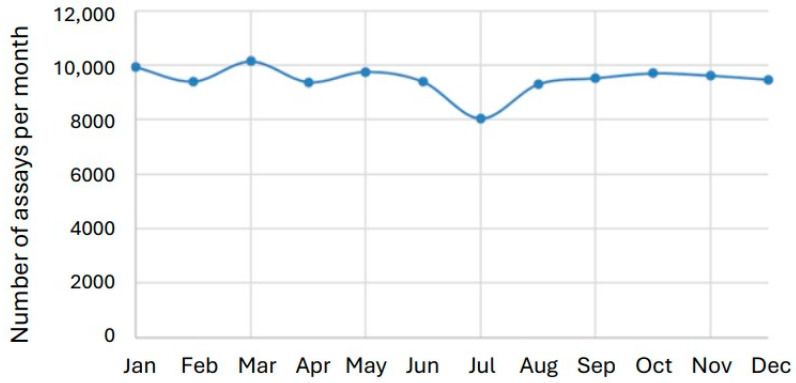
Number of tacrolimus assays/month. The data is presented for years 2006–2024.

**Figure 4 biomedicines-14-00094-f004:**
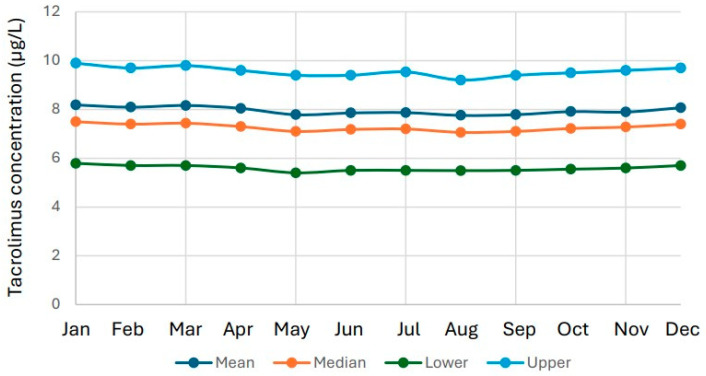
Tacrolimus results per month. The data is presented as 10th percentile, lower quartile, median, upper quartile and 90th percentile per year from 2006 to 2024.

**Figure 5 biomedicines-14-00094-f005:**
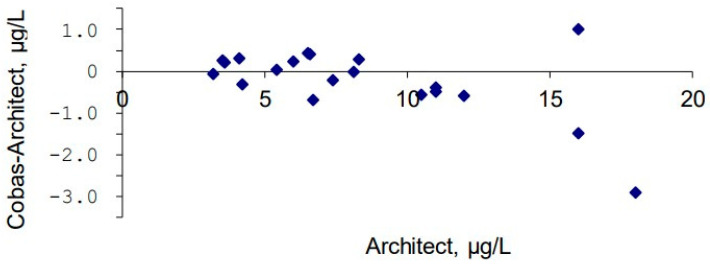
Bland–Altman plot showing patient results analyzed with the Architect method (reference method at that time, *x*-axis) vs. the difference between the Cobas and Architect methods (*y*-axis). The median difference between the two methods was 0.23 µg/L.

## Data Availability

The dataset used and analyzed during the current study is available from the corresponding author on reasonable request. The data are not publicly available due to ethical reasons as the ethical approval limits the use of the data.

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
