# Peer review of "Decreasing Tacrolimus Concentrations in Routine Therapeutic Drug Monitoring Data Indicate Adherence to Updated Therapeutic Goals"

_biomedicines, 2026, doi:10.3390/biomedicines14010094_

Round 1

Reviewer 1 Report

Comments and Suggestions for Authors

Recommendation: Major revision

  • The abstract should be revised to better separate objectives, methods, results, and conclusions, with inclusion of numerical outcomes to enhance clarity.
  • The introduction is comprehensive but overly detailed in pharmacological mechanisms, which could be condensed to strengthen focus on study rationale and objectives.
  • The main novelty lies in applying patient medians as a quality control tool, which is innovative but not fully developed or demonstrated within the paper.
  • The study is well organized and clearly written, but it mainly provides descriptive data without sufficient inferential statistical analysis to substantiate claims of “steady decline” or “no seasonal variation.”
  • Trend analysis should include statistical methods such as regression, Mann–Kendall, or other time-series tests to confirm significance rather than relying solely on visual inspection of graphs.
  • Figures lack adequate details—axes are unlabeled, and there are no error bars or statistical indicators. Each figure should be described more clearly, with proper scales, units, and legends to improve readability.
  • The decrease in tacrolimus concentrations is plausibly linked to updated therapeutic targets, but this interpretation remains speculative since no direct evidence or clinical context (e.g., transplant type, time since transplantation, concurrent therapies) is available.
  • The lack of clinical metadata is a major limitation that restricts understanding of patient-level or organ-specific patterns. Stratification by age or sex could partially address this limitation and add analytical depth.
  • Discussion of assay changes is adequate but should include quantitative evaluation of method transition effects; even minor shifts in calibration could influence long-term trends.
  • The idea of patient median monitoring is valuable but should be supported with at least one practical example where this approach identified, or could have identified, an analytical drift or reagent lot issue.
  • Table 1 is mentioned but not presented in the text provided; ensure it is included and clearly labeled with parameters and units.
  • The section on seasonal variation is descriptive and should include statistical testing to verify that no significant fluctuations exist.
  • The discussion tends to repeat background information; it could be made more concise and structured around key interpretations and implications.
  • Language quality is generally high, but minor grammatical corrections are required (“data were,” not “data was”) and unit formatting should be standardized (µg/L).
  • Ethical approval and data protection are appropriately addressed, though the justification for the waiver of consent could be explained earlier in the methods section.
  • Statistical robustness and figure quality are the main areas needing improvement to elevate this from a descriptive report to a rigorous analytical study.
  • The conclusions are consistent with results but should include quantitative data (e.g., “median values decreased by approximately 25–30%”) to emphasize the magnitude of change.

Author Response

Dear Editor

Thank you very much for the opportunity to improve our manuscript. We very much appreciate the valuable comments from the reviewer, that has significantly improved the manuscript. Below is a point by point response to the issues that the reviewer has raised.

Yours sincerely

Anders Larsson

Reviewer 1

  • The abstract should be revised to better separate objectives, methods, results, and conclusions, with inclusion of numerical outcomes to enhance clarity.

Revised abstract:

Background/Objectives:
Tacrolimus is a key immunosuppressive drug used to prevent organ rejection after transplantation. Its narrow therapeutic window and high interindividual pharmacokinetic variability make therapeutic drug monitoring (TDM) essential. This study aimed to (1) characterize long-term trends in tacrolimus concentrations, (2) assess potential seasonal variation, and (3) evaluate the suitability of patient medians as a tool for laboratory quality monitoring.

Methods:
A retrospective analysis was conducted on 113,735 tacrolimus whole-blood results obtained between 2006 and 2024 at Uppsala University Hospital, Sweden. Samples were analyzed using immunoassays on the Hitachi 912 (Microgenics) until 2008, the Abbott Architect until 2021, and the Roche Cobas Pro e 801 thereafter. Annual patient medians and percentiles (10th, 25th, 50th, 75th, 90th) were calculated. Seasonal variation was assessed by comparing monthly test volumes and concentration distributions.

Results:
The annual number of tacrolimus results increased from 5,616 in 2006 to 7,320 in 2024, comprising 67,133 male and 46,602 female patient results. All distribution metrics declined steadily over the study period, with median tacrolimus concentrations decreasing by 20–30%. July test volume was approximately 15% lower than other months, but no meaningful seasonal variation in tacrolimus concentrations was observed; monthly medians and percentiles remained stable throughout the year.

Conclusions:
Tacrolimus concentrations at the population level have decreased consistently over nearly two decades. These findings likely reflect evolving clinical practice, including dose minimization strategies to reduce toxicity. Patient medians proved robust over time and may serve as a useful adjunct to conventional quality control, particularly when commercial control materials do not fully mimic patient sample behavior.

  • The introduction is comprehensive but overly detailed in pharmacological mechanisms, which could be condensed to strengthen focus on study rationale and objectives.

The version that we submitted originally was shorter (both introduction and discussion), but the editor specifically requested that we added more text. Thus, there is a disagreement on the recommended length of the introduction between editor and reviewer. We have therefore refrained from shortening the introduction. If the editor wants us to shorten it we will do so.

  • The main novelty lies in applying patient medians as a quality control tool, which is innovative but not fully developed or demonstrated within the paper.

We have now added statistical analysis of the changes over time.

  • The study is well organized and clearly written, but it mainly provides descriptive data without sufficient inferential statistical analysis to substantiate claims of “steady decline” or “no seasonal variation.”

We have now added Spearman rank analysis for females and males separately  showing a highly significant association but weak Spearman rank values for both males and females.

Over the study period, the number of results increased from 5,616 in 2006 to 7,320 in 2024, comprising a total of 46,602 results from female and 67,133 results from male patients. The median age of the female study subjects was 52 years (interquartile range (IQR) 39-63 years) at the time of their testing and they had a mean tacrolimus value of 7.99 ug/L (IQR 5.60-9.60). The female test results showed a significant decrease in tacrolimus values over time (Spearman rank -0.214; p<0.00000000001). There was also a very weak positive association between age and tacrolimus results (Spearman rank 0.030; p<0.00000001). The median age of the male study subjects was 52 years (interquartile range (IQR) 42-66 years) at the time of their testing and they had a mean tacrolimus value of 7.96 ug/L (IQR 5.60-9.60). The male test results showed a significant decrease in tacrolimus values over time (Spearman rank -0.190; p<0.00000000001). There was also a very weak positive association between age and tacrolimus results (Spearman rank 0.058; p<0.00000000001).

  • Trend analysis should include statistical methods such as regression, Mann–Kendall, or other time-series tests to confirm significance rather than relying solely on visual inspection of graphs.

We have performed regression analysis for females and males separately to verify that the pattern was similar in both males and females.

  • Figures lack adequate details—axes are unlabeled, and there are no error bars or statistical indicators. Each figure should be described more clearly, with proper scales, units, and legends to improve readability.

The figures have now been revised. The statistical analysis is presented together with the figures in the results section.

  • The decrease in tacrolimus concentrations is plausibly linked to updated therapeutic targets, but this interpretation remains speculative since no direct evidence or clinical context (e.g., transplant type, time since transplantation, concurrent therapies) is available.

Adding clinical data would be a direct violation of the ethical permit. As specified in the manuscript only age and gender of the study subjects were allowed. The study covers a time period of 20 years. This means that a substantial number of participants most likely have died which makes it impossible to obtain informed consent from each of the 113,000 patient which would be required for adding additional clinical data.

  • The lack of clinical metadata is a major limitation that restricts understanding of patient-level or organ-specific patterns. Stratification by age or sex could partially address this limitation and add analytical depth.

We have now added associations between age and tacrolimus values and median and interquartile ranges for both males and females. There is also spearman rank testing performed for each sex.

  • Discussion of assay changes is adequate but should include quantitative evaluation of method transition effects; even minor shifts in calibration could influence long-term trends.

We fully agree with the reviewer comment. We now have added an evaluation of the method transition effects providing data from the method comparison.

  • The idea of patient median monitoring is valuable but should be supported with at least one practical example where this approach identified, or could have identified, an analytical drift or reagent lot issue.

We recently published a study showing a clear shift of the Roche folate method using patient median values (Larsson A, Saldeen J, Duell F. Recent decline in patient serum folate test levels using Roche Diagnostics Folate III assay. Clin Chem Lab Med. 2025 Aug 14;63(12):e275-e277.) We sent the folate manuscript to Roche and informed them that it was submitted to CCLM. Almost simultaneously as the article was published, Roche changed their calibration.

Added to the discussion: We recently showed a clear shift of the Roche folate method using patient median values (Larsson A, Saldeen J, Duell F. Recent decline in patient serum folate test levels using Roche Diagnostics Folate III assay. Clin Chem Lab Med. 2025 Aug 14;63(12):e275-e277.) supporting the use of patient medians to detect analytical drift.

  • Table 1 is mentioned but not presented in the text provided; ensure it is included and clearly labeled with parameters and units.

We have now removed the mentioning of table 1 as the information in that table is covered in Figures 3 and 4. We apologize for this error.

  • The section on seasonal variation is descriptive and should include statistical testing to verify that no significant fluctuations exist.

We have added a Mann-Whitney analysis comparing July, that had the lowest number of requests with May and September.

The number of Tacrolimus results during July was 8041 while 19299 for May plus September, thus a reduction of 16.7% in July. Despite this reduction the median tacrolimus result was 7.2 ug/L in July while 7.1 ug/L in May+August. Thus despite a clear reduction in number of test results the median tacrolimus result only increased by 1.4%.

  • The discussion tends to repeat background information; it could be made more concise and structured around key interpretations and implications.

The version that we submitted originally was shorter (both introduction and discussion) but the editor specifically requested that we should add more text. Thus, there is a disagreement on the recommended length of the discussion between editor and reviewer. We have therefore refrained from shortening the introduction. If the editor wants us to shorten it we will do so.

  • Language quality is generally high, but minor grammatical corrections are required (“data were,” not “data was”) and unit formatting should be standardized (µg/L).

We have revised the language.

  • Ethical approval and data protection are appropriately addressed, though the justification for the waiver of consent could be explained earlier in the methods section.

Added: Because the results were reported only at the group level, there was no risk of identifying individual patients, and informed consent was therefore not required by the ethics committee. Furthermore, all testing was performed as part of the patients’ routine clinical workup, meaning that no additional blood sampling was necessary for the study.

  • Statistical robustness and figure quality are the main areas needing improvement to elevate this from a descriptive report to a rigorous analytical study.

Addition of statistical analysis of changes over time and seasonal variation.

  • The conclusions are consistent with results but should include quantitative data (e.g., “median values decreased by approximately 25–30%”) to emphasize the magnitude of change.

Addition of statistical analysis of changes over time and seasonal variation. 

Reviewer 2

This study analyzed more than 113000 whole blood samples containing  tacrolimus from patients undergo TDM  using rigorous longitudinal retrospective methods  (analytical platforms) from Hitachi 912 via Abbott Architect to Roche Cobas Pro controlled by the validation procedure. A limiting characteristic of Tacrolimus is the high intra and interpatient variability associated with its use. TDM  is necessary  to facilitate  Tac management and  to avoid undesirable clinical outcomes

But, I have some comments and suggestions to the authors:

  1. The materials and methods section should be improved. The authors should provide information regarding time of extraction, type of transplant, post-transplant time to which each determinant corresponds, age of patients. In the other words, demographic data of patients.

Adding clinical data would be a direct violation of the ethical permit. As specified in the manuscript only age and gender of the study subjects were allowed. The study covers a time period of 20 years. This means that a substantial number of participants most likely have died which makes it impossible to obtain informed consent from each of the 113,000 patient which would be required for adding additional clinical data.

  1. The Results section should be reevaluated: how many samples correspond to men and women, and how many correspond to the same person, Tac concentrations were in steady-state or not.

We have added information on the number of samples from men and women and also how many study subjects that were included.

  1. It is very important to include details concerning bias mitigation during platform transition over the years. Methodological changes need to be validated (Passing-Bablok regression, Bland-Altman analysis). Among limitations of the study there are other multiple causes for the decreases in the amount of samples f.e. patient mortality and relocation, technical  problems with equipment etc.  Have the authors evaluated such things?

Adding clinical data would be a direct violation of the ethical permit. As

  1. In the discussion the absence of seasonal fluctuations in concentrations measure were discussed. I am not sure, if authors discussed such fluctuations in the submitted manuscript.

We have added a Mann-Whitney analysis comparing July, that had the lowest number of requests with May and September.

The number of Tacrolimus results during July was 8041 while 19299 for May plus September, thus a reduction of 16.7% in July. Despite this reduction the median tacrolimus result was 7.2 ug/L in July while 7.1 ug/L in May+August. Thus, despite a clear reduction in number of test results the median tacrolimus result only increased by 1.4%.

  1. There is also no information about tacrolimus formulation that was administered to patients.

Adding clinical data would be a direct violation of the ethical permit.

  1. Please describe the selection criteria for data distribution testing methods (such as Shapiro Wilk test) and non-parametric methods.

Over the study period, the number of results increased from 5,616 in 2006 to 7,320 in 2024, comprising a total of 46,602 results from female and 67,133 results from male patients. The median age of the female study subjects was 52 years (interquartile range (IQR) 39-63 years) at the time of their testing and they had a mean tacrolimus value of 7.99 ug/L (IQR 5.60-9.60). The female test results showed a significant decrease in tacrolimus values over time (Spearman rank -0.214; p<0.00000000001). There was also a very weak positive association between age and tacrolimus results (Spearman rank 0.030; p<0.00000001). The median age of the male study subjects was 52 years (interquartile range (IQR) 42-66 years) at the time of their testing and they had a mean tacrolimus value of 7.96 ug/L (IQR 5.60-9.60). The male test results showed a significant decrease in tacrolimus values over time (Spearman rank -0.190; p<0.00000000001). There was also a very weak positive association between age and tacrolimus results (Spearman rank 0.058; p<0.00000000001).

Reviewer 2 Report

Comments and Suggestions for Authors

This study analyzed more than 113000 whole blood samples containing  tacrolimus from patients undergo TDM  using rigorous longitudinal retrospective methods  (analytical platforms) from Hitachi 912 via Abbott Architect to Roche Cobas Pro controlled by the validation procedure. A limiting characteristic of Tacrolimus is the high intra and interpatient variability associated with its use. TDM  is necessary  to facilitate  Tac management and  to avoid undesirable clinical outcomes

But, I have some comments and suggestions to the authors:

  1. The materials and methods section should be improved. The authors should provide information regarding time of extraction, type of transplant, post-transplant time to which each determinant corresponds, age of patients. In the other words, demographic data of patients.
  2. The Results section should be reevaluated: how many samples correspond to men and women, and how many correspond to the same person, Tac concentrations were in steady-state or not.
  3. It is very important to include details concerning bias mitigation during platform transition over the years. Methodological changes need to be validated (Passing-Bablok regression, Bland-Altman analysis). Among limitations of the study there are other multiple causes for the decreases in the amount of samples f.e. patient mortality and relocation, technical  problems with equipment etc.  Have the authors evaluated such things?
  4. In the discussion the absence of seasonal fluctuations in concentrations measure were discussed. I am not sure, if authors discussed such fluctuations in the submitted manuscript.
  5. There is also no information about tacrolimus formulation that was administered to patients.
  6. Please describe the selection criteria for data distribution testing methods (such as Shapiro Wilk test) and non-parametric methods.

Author Response

(The authors gave the same response as above.)
